# Selected Mediators of Inflammation in Patients with Acute Ischemic Stroke

**DOI:** 10.3390/ijms231810614

**Published:** 2022-09-13

**Authors:** Hanna Pawluk, Renata Kołodziejska, Grzegorz Grześk, Mariusz Kozakiewicz, Alina Woźniak, Mateusz Pawluk, Agnieszka Kosinska, Magdalena Grześk, Jakub Wojtasik, Grzegorz Kozera

**Affiliations:** 1Department of Medical Biology and Biochemistry, Faculty of Medicine, Collegium Medicum in Bydgoszcz, Nicolaus Copernicus University in Toruń, Karłowicza 24, 85–092 Bydgoszcz, Poland; 2Department of Cardiology and Clinical Pharmacology, Faculty of Health Sciences, Collegium Medicum in Bydgoszcz, Nicolaus Copernicus University in Toruń, Ujejskiego 75, 85-168 Bydgoszcz, Poland; 3Division of Biochemistry and Biogerontology, Department of Geriatrics, Faculty of Health Sciences, Collegium Medicum in Bydgoszcz, Nicolaus Copernicus University in Toruń, Dębowa 3, 85-626 Bydgoszcz, Poland; 4Centre for Languages & International Education, University College London, 26 Bedford Way, London WC1H 0AP, UK; 5Statistical Analysis Centre, Nicolaus Copernicus University in Toruń, Chopin 12/18, 87-100 Toruń, Poland; 6Medical Stimulation Centre, Medical University of Gdańsk, Dębowa 25, 80-204 Gdańsk, Poland

**Keywords:** ischemic stroke, interleukin-6, tumor necrosis factor, modified Rankin Scale, National Institutes of Health Stroke Scale

## Abstract

During a stroke, a series of biochemical and metabolic changes occur which eventually lead to the death of cells by necrosis or apoptosis. This is a multi-stage process involving oxidative stress and an inflammatory response from the first signs of occlusion of a blood vessel until the late stages of regeneration and healing of ischemic tissues. The purpose of the research was to assess the concentration of pro-inflammatory cytokines IL-6 and TNF-α in the blood serum of patients with ischemic stroke (AIS) and to investigate their role as new markers in predicting functional prognosis after thrombolytic therapy. The researches have shown that the concentrations of the measured biomarkers were higher compared to the control group. Serum levels of IL-6 and THF-α before the initiation of intravenous thrombolysis were lower in the subgroup of patients with a favourable functional result (mRS: 0–2 pts) compared to the group of patients with an unfavourable functional result (mRS: 3–6 pts). A positive correlation was found between the concentration of IL-6 and TNF-α in patients with AIS during <4.5 h and on one day after the onset of stroke, which means that the concentration of IL-6 increases with the increase in TNF-α concentration. It has also been shown that higher levels of IL-6 in the acute phase of stroke and on the first and seventh days, and TNF-α during onset, were associated with poorer early and late prognosis in patients treated with intravenous thrombolysis. A relationship was found between the level of IL-6 and TNF-α in the subacute AIS and the severity of the neurological deficit. It has been shown that the investigated biomarkers may be a prognostic factor in the treatment of thrombolytic AIS.

## 1. Introduction

Stroke is the second leading cause of death and permanent disability, after the heart attack, in people above forty. The origins of the stroke could be related to heart embolism, cerebral microangiopathy, haematological diseases, as well as extracranial and intracranial arteriosclerosis [1,2,3,4]. During a stroke, there is a risk of rupturing blood vessels and bleeding in some parts of the brain. Therefore, the strokes can be classified as ischemic or haemorrhagic, which comprise about 85% and 15% of the reported cases, respectively [3]. Patients who survive the acute phase of the disease require intensive care due to physical, verbal, and intellectual disability. The neurological scale is employed as a tool to measure the degree of damage caused by a stroke.

During a stroke, a lack of blood flow in the ischemic core of the brain is responsible for a decrease in the oxygen, glucose, and lipids supplied to the brain neurons and other neighbouring cells and tissues. This causes several different biochemical and metabolic changes, which eventually lead to the death of cells by necrosis or apoptosis [5,6,7,8]. This is a multi-layered, complex, and surging process which involves oxidative stress and inflammation from the first signs of occlusion of a blood vessel until the late stages of regeneration and healing of ischemic tissues [5,6,7]. Consequently, the microvessels are damaged as well as glial, and neuronal cells are activated. In addition, a dysfunction in the brain–blood barrier (BBB) and leukocyte infiltration is observed. Brain damage not only increases the inflammation in the ischemic brain but is also responsible for the systemic immunological reaction concurrent with lymphocytopenia that causes immunodepression [6]. However, the immunological response of the body remains a controversial issue and the subject of many arguments, especially whether the role of the immune system carries beneficial or harmful consequences [9]. There is an opinion that the immune system can act as a promising and important tool employed in the modulation of brain injury during and after a stroke.

However, despite the extensive research, no significant indicators (markers) have been selected as the ones which could successfully prevent, detect, and treat strokes. It has been proven that excitotoxicity and an increase in the reactive oxygen species (ROS) cause an expression of many pro-inflammatory genes by the excitation of various transcription factors. In addition, the activated microglia and astrocytes give rise to the secretion of pro-inflammatory cytokines, anti-inflammatory chemokines, and matrix metalloproteinases (MMP) [7,9,10]. At the surface of the endothelium, inflammatory mediators induce the expression of cell adhesion that mediates the adherence of leukocytes to the lesion site [6]. The influence of inflammatory bio-markers on the long-term prognosis of patients treated with rt-PA (recombinant tissue plasminogen activator) employed in intravenous thrombolytic and endovascular therapy remains an object of controversy. It has been suggested that molecules that affect a patient’s prognosis are cytokines such as interleukin-1 (IL-1), interleukin-6 (IL-6) and tumour necrosis factor-alpha (TNF-α). However, there is no clear information related to their concentrations and their overall effect on cerebral infarction.

This research aims to illustrate the involvement of inflammation biomarkers such as IL-6 and TNF-α in the aetiology of acute ischemic stroke. The data was collected from the group of patients at different time intervals, namely < 4.5 h on the first and seventh day since the stroke, and compared with the results obtained from the group of healthy participants (control group) which allowed for the evaluation of those inflammation biomarkers as novel factors used in predicting functional prognosis after thrombolytic therapy. Careful analysis of these parameters will also help better comprehend how inflammation contributes to the progression of stroke in cerebral circulation disorders. This can be recognised as a valuable indication for modulating immune response mechanisms to limit damage to nervous tissue following ischemia.

## 2. Results

A total of 125 patients took part in this study. Data were collected from 124 participants at the point of hospital admission, 121 at the point of discharge from the hospital, 113 after 3 months, and 91 after a year. The causes of death in patients after stroke were neurological—cerebral disorders (cerebral haemorrhage, cerebral edema) and extracerebral (severe circulatory and respiratory failure, pneumonia, sepsis, and other infections).

A favourable outcome after intravenous thrombolysis, defined as 0–2 points on the modified Rankin Scale (mRS) scale, was observed on admission to the hospital in 98 (79%), on discharge in 91 (75%), after 3 months in 85 (75%), and after 1 year in 79 (86.8%) patients. In total, 15 patients died, 8 of them in the first 3 months since the stroke. On admission, hypertension was detected in 103 (83.1%) patients, diabetes mellitus was found in 52 (41.9%), coronary artery disease in 27 (21.8%), atrial fibrillation (chronic or paroxysmal) in 15 (12.1%), hyperlipidaemia in 13 (10.5%), chronic kidney disease in 3 (2.4%), and hyperuricemia in 6 (4.8%). Before the onset of stroke, 36 (29.0%) patients were taking statins, 42 (33.9%) were taking antiplatelet drugs, and 3 (2.4%) were receiving anticoagulant treatment. Patients who benefited from the positive effect of the treatment suffered less from atrial fibrillation (*p* = 0.040) and coronary artery disease (*p* = 0.010). They also less frequently took antiplatelet drugs (*p* = 0.002), and less frequently experienced infections (*p* = 0.007). Patients who demonstrated favourable results were younger (*p* = 0.003) and had relatively small neurological changes based on the NIHSS scale at admission (*p* < 0.001), at discharge (*p* < 0.001) and based on the mRS scale at discharge (*p* < 0.001), at months (*p* < 0.001) and at 1 year (*p* = 0.002), as illustrated by Table 1.

The study assessed the concentration of cytokines at times < 4.5 h (0), 24 h (1) and on the seventh day (2). The mean concentration of cytokines was calculated, taking into account the change in the number of patients (admission, discharge, three months, one year). The general study design is illustrated in Figure 1.

Serum IL-6 concentrations were measured before the intravenous thrombolysis treatment. The results indicated that serum IL-6 levels were lower in the subgroup of patients with a favourable functional result (mRS: 0–2 pts) compared to the group of patients with an unfavourable functional result (mRS: 3–6 pts) assessed at admission, discharge, after 3 months, and after 1 year. Levels of this biomarker < 4.5 h were 4.69 with IQR 3.69–6.64 vs. 6.48 with IQR 5.81–8.70 pg/mL (*p* = 0.012); 5.92 with IQR 3.84–6.52 vs. 10.0 with IQR 7.38–11.6 pg/mL (*p* < 0.001); 5.82 with IQR 3.79–6.51 vs. 10.8 with IQR 7.23–13.5 pg/mL (*p* < 0.001); 5.92 with IQR 3.87–6.57 vs. 12.4 with IQR 11.1–14.6 pg/mL (*p* < 0.001), respectively. There were statistically significant differences observed in the IL-6 levels marked after 24 h in the case of patients with favourable (mRS: 0–2 pts) and unfavourable (mRS: 3–6 pts) functional results at the time of admission, discharge, and after 3 months. Similar statistical outcomes were reported in IL-6 concentrations determined after 7 days on admission. IL-6 levels after 24 h were 6.55 with IQR 6.48–6.79 vs. 7.68 with IQR 6.51–11.6 pg/mL (*p* = 0.024); 6.79 with IQR 6.36–7.64 vs. 12.1 with IQR 9.71–14.1 pg/mL (*p* < 0.001); 6.79 with IQR 6.37–7.79 vs. 12.6 with IQR 9.30–15.0 pg/mL (*p* = 0.037), while the IL-6 concentrations after 7 days were 4.71 with IQR 3.72–5.49 vs. 6.87 with IQR 4.74–7.69 pg/mL (*p* = 0.022), respectively. Moreover, TNF-α biomarker levels < 4.5 h in patients with mRS: 0–2 pts in comparison to TNF-α levels in patients with mRS: 3–6 pts at the time of being discharged from the hospital, after 3 months and after 1 year, also resulted in significant statistical differences. Namely, TNF-α biomarker concentrations < 4.5 h were 36.4 with IQR 31.0–49.0 vs. 50.2 with IQR 45.0–52.5 pg/mL (*p* = 0.010); 33.9 with IQR 31.0–46.5 vs. 51.8 with IQR 48.6–59.6 pg/mL (*p* < 0.001); 33.9 with IQR 30.9–46.5 vs. 52.5 with IQR 49.1–56.0 pg/mL (*p* = 0.006), respectively. In addition, there were considerable statistical variations in data collected after 24 h (43.3 with IQR 34.1–50.0 vs. 68.4 with IQR 64.6–72.3 pg/mL z *p* = 0.017) in patients a year from the stroke, that are summarised in Table 2.

The obtained results of IL-6 levels, determined in the time interval < 4.5 h, on day 1 and day 7, were statistically higher in comparison to the concentration of IL-6 in the control group (*p* < 0.001); they were 6.37, 7.09 and 6.54 vs. 3.59 pg/mL. Furthermore, the numerically significant differences in the concentration of TNF-α (42.4, 46.2, 33.9 vs. 29.8 pg/mL) concerning the control group were also observed, as shown in Table 3.

Correlations between the concentration of IL-6 measured in the blood serum during the onset and on the 1st day of ischemic stroke and the NIHSS results assessed on admission (R = 0.43, *p* < 0.01 and R = 0.4, *p* < 0.01, respectively) and on discharge (R = 0.61, *p* < 0.01 and R = 0.52, *p* < 0.01, respectively) were seen (Table 4). Additionally, correlations between IL-6 levels determined in the blood serum in < 4.5 h and in the 1st day of ischemic stroke and results evaluated according to the mRS scale on admission (R = 0.52, *p* < 0.01 and R = 0.44, *p* < 0.01, respectively), on discharge (R = 0.61, *p* < 0.01 and R = 0.47, *p* < 0.01, respectively), after 3 months (R = 0.68, *p* < 0.01 and R = 0.3, *p* < 0.01, respectively) and 1 year since the stroke (R = 0.73, *p* < 0.01 i R = 0.50, *p* < 0.01, respectively) and the concentrations of IL-6 measured after 7 days on discharge (R = 0.28, *p* = 0.04) were also observed. Moreover, the relationship between the TNF-α levels measured during onset and the NIHSS scale on admission (R = 0.4, *p* = 0.02) as well as mRS values assessed on admission (R = 0.33, *p* = 0.01) and in the 3rd month since the stroke (R = 0.47, *p* < 0.01).

The additional studies revealed the correlations between the concentrations of IL-6 and TNF-α measured in various time intervals. The significant statistical connection between IL-6 levels during onset and TNF-α levels assessed in < 4.5 h and after 24 h since the stroke (*p* < 0.01) were detected as illustrated by Figure 1.

There was no evidence of the influence of statins and age on the levels of the studied biomarkers.

The concentrations of IL-6 in the blood serum during onset demonstrated good sensitivity and specificity in predicting patients’ functional outcomes, both determined after discharge from hospital (cut-off point, IL-6 = 7.11 pg/mL, sensitivity = 88.6%, specificity = 83.3%, AUC = 0.887), during 90 days (cut-off point, IL-6 = 6.54 pg/mL, sensitivity = 77.3%, specificity = 93.3%, AUC = 0.918) as well as 1 year (cut-off point, IL-6 = 8.55 pg/mL, sensitivity = 95.2%, specificity = 100%, AUC = 0.992) since the stroke. Similar results were observed for IL-6 levels assessed in the 1st day since the stroke on discharge (cut-off point, IL-6 = 8.22 pg/mL, sensitivity = 85.7%, specificity = 100%, AUC = 0.921), after 90 days (cut-off point, IL-6 = 8.94 pg/mL, sensitivity = 78.1%, specificity = 75%, AUC = 0.731) as well as 1 year (cut-off point, IL-6 = 8.94 pg/mL, sensitivity = 78.1%, specificity = 75%, AUC = 0.777) since the stroke. The summary of the results is illustrated by Figure 2.

Good sensitivity and specificity were also found in the TNF-α levels’ assessment <4.5 h and on the 1st day since the stroke during evaluation of the patients’ neurologic disability; after 3 months (<4.5 h: cut-off point = 39.94 pg/mL, sensitivity = 62.8%, specificity = 100%, AUC = 0.840; after a day: cut-off point = 60.14 pg/mL, sensitivity = 96.7%, specificity = 50.0%, AUC = 0.733) and 1 year (<4.5 h: cut-off point = 42.54 pg/mL, sensitivity = 62.8%, specificity = 100%, AUC = 0.832; after a day: cut-off point = 60.14 pg/mL, sensitivity = 60.14%, specificity = 100%, AUC = 0.966) since the stroke. The outline of the results is summarised by Figure 3.

Additionally, the analysis of inflammatory biomarkers was done for different types of strokes. The IL-6 (5.59 pg/mL vs. 7.29 pg/mL) with *p* = 0.016 and TNF-α levels (33.43 pg/mL vs. 49.60 pg/mL) with p = 0.017 assessed in time < 4.5 h were statistically lower in patients suffering from the LACI (lacunar cerebral infarct) ischemic stroke in comparison to the patients with PACI (partial anterior circulation infarct) stroke. Moreover, the statistical differences were reported for LACI and POCI (posterior circulation infarction) concerning TNF-α assessed during onset (33.43 pg/mL vs. 48.82 pg/mL z *p* = 0.04) and on the 1st day (37.24 pg/mL vs. 48.86 pg/mL) with p = 0.018 since the stroke, as summarized in Table 5 and illustrated by Figure 4.

The correlation between the concentration of biomarkers and mortality was also evaluated. The increased mortality of patients with AIS was observed for the concentration of TNF-α determined within 1 day after exceeding the value of 50 pg/mL. The rise in mortality was also seen for the IL-6 and TNF-α levels measured on the 7th day (>4 pg/mL for IL-6 and >37 pg/mL for TNF-α) as presented in Figure 5.

The summary of the study is presented in Figure 2.

## 3. Discussion

The neuroimmune axis works based on a feedback loop, in which ischemic foci of the brain cause a wide activation of inflammatory cytokines in peripheral immune systems, which in turn modulate the pathophysiology of the central nervous system (CNS) [11].

The mechanism of stroke is a multistep pathological process, and the maintenance of the neuroimmune balance has a significant impact on patients’ prognoses. Therefore, the activity of pro-inflammatory cytokines may be important for the effectiveness of therapy in patients with arterial ischemic stroke (AIS).

In this study, a higher concentration of inflammatory biomarkers IL-6, and TNF-α within 4.5, 24 h and 7 days after the onset of stroke was observed compared to healthy participants who acted as a control group. However, the highest levels of inflammatory cytokines were observed within the first 24 h from the appearance of clinical changes in stroke.

Our findings have been confirmed by other researchers who believe that IL-6 and TNF-α may be the key inflammatory markers in stroke patients, showing a significant increase in serum levels in numerous studies, within a few hours after the onset of ischemia [12,13,14,15] up to the 90th day after stroke [16,17,18,19,20]. Tuttolomondo et al. reported that elevated levels of TNF-α after stroke facilitated several biochemical processes. One of these processes involved the expression of tissue factors and the adhesion of molecules for leukocytes, as well as the release of interleukin-1 (IL-1), nitric oxide, and factor VIII/von Willebrand. In addition, the platelet-activating factor and endothelin were also expressed. Furthermore, the suppression of the thrombomodulin–protein S–protein C has been noticed, as well as the reduction of tissue plasminogen activator and release of plasminogen activator-1 inhibitor. All these biochemical changes may indicate a pro-inflammatory role of TNF-α in the central nervous system [21].

The cytokines TNF-α and IL-6 modulated infarct evolution in the experimental stroke and therefore received much attention as putative markers of stroke severity and neurological outcomes in humans. It has been found that even a small increase in biomarkers can have a significant effect on the evolution of infarcts if released at the site of ischemic neurons. Many authors indicate that the rise in the concentration of IL-6 in the blood serum on the first day after a stroke is associated with the deterioration of the functional status [16,22,23,24] of patients and a greater extent of the ischemic focus [13,23]. Initial levels of IL-6 have been shown to correlate with the severity of stroke (size of the lesion), perfusion-weighted imaging (PWI) signal, and the patient’s neurological score [25,26,27]. Sotgiu et al. observed, in 66 patients with stroke, an inverse relationship between the early IL-6 levels, the size of the lesion and the neurological result of the patient, which indicates that initial IL-6 has a neuroprotective effect and is not a marker of disease progression [28].

There are also conflicting reports of stroke-induced changes in TNF-α levels in serum and plasma due to its complex and pleiotropic signalling nature. It has been shown that the increase in TNF-α concentration within 24 and 48 h after stroke, and the observed slow decrease occurring within 72 and 144 h after stroke, correlates with clinical improvement in patients in the acute phase of ischemic stroke [20]. Also, others report elevated TNF-α levels, although with different peak times and contradictory results concerning the correlation between peripheral TNF-α levels and stroke severity [29,30]. Therefore, TNF-α is a sensitive parameter determining the onset of the inflammatory response and can be used as a useful prognostic indicator [31,32]. However, some studies have shown that TNF-α levels remain unchanged after stroke [25,26,33].

A growing body of evidence shows, in experimental models as well as in clinical relationships, that neurological inflammation promotes further injury that causes cell death. Inflammation contributes to oedema and may facilitate apoptosis. Cytokines are primarily responsible for the onset of the ischemic inflammatory cascade and are involved in the progression of cerebral infarction and may influence the severity and outcome of the disease. However, it should be considered that TNF-α and IL-6 also play a role in protecting the brain parenchyma against damage caused by ischemic stroke in the subsequent ischemic stages. Consequently, these biomarkers appear to have a dual effect on encephalitis that follows the early stages of ischemic brain injury. They play a pro-inflammatory role during the first step of inflammation in the central nervous system and an immunosuppressive and neurotropic role in the chronic phase [7,8].

In our study, we have demonstrated that serum levels of IL-6 before the intravenous thrombolysis were lower in the subgroup of patients with a favourable functional outcome compared to the group of patients with an unfavourable functional result (mRS: 3–6 pts) assessed on admission, on discharge, after 3 months and after a year since the stroke. There were also statistically significant differences in IL-6 concentrations assessed after 24 h for patients with mRS: 0–2 pts and mRS: 3–6 pts both at admission, discharge and after three months, and after 7 days only at admission. Moreover, there were also statistically important differences in the concentration of the TNF-α biomarker < 4.5 h in the subgroup of patients with a favourable functional result in comparison to patients with an unfavourable result at the time of discharge from the hospital, after three months and after a year, and TNF-α levels assessed after 24 h only after a year (Table 2). It was also shown that a higher IL-6 concentration in the acute phase of stroke and assessed on day 1 and day 7 only at discharge was associated with poorer initial and late prognosis in patients treated with intravenous thrombolysis. A correlation was found between TNF-α levels measured during the onset time, with the NIHSS scale on admission, and mRS assessed in the third month after stroke (Table 4). A relationship between the levels of IL-6 and TNF-α and the severity of the neurological deficit was observed. We have therefore illustrated that the studied biomarkers could act as prognostic factors in the treatment of thrombolytic AIS. Similarly, other researchers have shown that high plasma levels of IL-6 in post-stroke patients have been correlated with poor prognosis [19,24,34] and expansive ischemic lesions [13,34,35]. In addition, increased plasma levels of TNF-α determined on admission also correlated with the volume and severity of the stroke and the functional outcome of the patient [24,31,32]. Nevertheless, there are reports in the literature which do not confirm such relationships [36,37].

In turn, Welsh et al. demonstrated a relationship between the level of TNF-α and the risk of recurrent infarction and emphasized that it occurs only in people with very high concentrations of this marker [38].

Since we have shown a significant association between IL-6 and TNF-α and functional outcomes in a group of patients without prior infection and other chronic inflammatory diseases, we can conclude that IL-6 and TNF-α are inflammatory markers of acutely ischemic brain injury. Their levels are not a result of chronic inflammation. In previous reports by other authors [22,39], the increase in IL-6 levels was often associated with poor functional outcomes, but it was not clear whether this was an independent effect or related to a co-infection.

Interestingly, we found a positive correlation between the concentration of IL-6 and TNF-α in patients with AIS assessed in <4.5 h and 1 day after the stroke, which means that the concentration of IL-6 rises with the increase in TNF-α concentration.

TNF-α stimulates the brain pericytes to increase the synthesis of IL-6 by activating the kappa B (NF-κB) factor [6,40]. The soluble form (sTNF-α) has also been shown to act at the systemic level, enhancing the phagocytic and cytotoxic effects of macrophages and the expression of cytokines such as IL-6 and IL-1. The peripheral production of TNF-α is observed not only in animal studies but also in clinical cases [7]. This is related to the pronounced pro-inflammatory changes in the brain resulting from focal ischemia, resulting in a broad activation of inflammatory cytokines in peripheral organs of the immune system.

By analysing inflammatory biomarkers between different types of strokes, we found statistically lower levels of IL-6 in patients with LACI-type AIS compared to PACI at <4.5 h. Statistical differences were also found between LACI and POCI for TNF-α assessed in <4.5 h and 1 day after the stroke. Similarly to other authors, we have demonstrated lower serum concentrations of IL-6 and TNF-α in patients with lacunar stroke caused by cerebral microangiopathy than in patients with other AIS aetiology [14,41,42,43]. This circumstance may possibly be due to the fact that the pathophysiology, prognosis, and clinical features of lacunar strokes are different from all other cerebral infarcts [44].

Licatai et al., while examining the level of cytokines depending on the stroke subtype, found that patients with cardio-embolic stroke revealed significantly higher levels of TNF-α and IL-6 in plasma compared to others. This was associated with an increased neurological deficit at admission assessed on the SSS scale and a higher degree of the acute phase of immuno-inflammatory activation compared to patients with lacunar stroke [12].

The correlation found between the concentration of IL-6 measured <4.5 h and 1 day after the onset of worsening of stroke symptoms (NIHSS scale), both on admission and discharge from the hospital, and TNF-α, and the NIHSS scale on admission proves that these cytokines can be considered a potential marker of the extent of ischemic brain injury and its further clinical consequences [45].

Moreover, A. Tuttolomondo et al. stated that patients with lacunar stroke without a change detectable in CT or MRI of the brain do not show any significant difference in the level of plasma cytokines in the acute phase compared to patients with the lacunar lesion. Thus, in addition to the direct relationship between infarct size and cytokine levels, the site of infarction may also play an additional role in acute-phase immuno-inflammatory activation [14].

The inflammatory cascade, which persists and continues throughout the stroke process, deteriorates functional status, increasing patient mortality. We have shown that serum IL-6 concentration above 4 pg/mL and TNF-α > 37 pg/mL, determined on the 7th day after stroke, may indicate a high probability of death in patients (Figure 5). These are novel data. Previously, the authors showed that the IL-6 level of 6.470 pg/mL is the threshold above which the survival rate of patients with acute stroke is reduced [19].

Inflammatory factors play a dual role in stroke and their mechanisms are complex. TNF-α and IL-6 are major regulators of the immune system and perform an important part in the spread of inflammation [46]. They are also central mediators in the immune processes of infection control, autoimmunity, allergic diseases, and anti-cancer activity [47].

Literature data in an animal model show that the administration of anti-TNF-α antibodies or TNF-α-binding protein (TNF-bp, soluble sTNF-R1, sTNF-p55R) has a neuroprotective effect. In preclinical studies, treatment with a TNF-α receptor antagonist (R-7050) also protected against neurological deficits after stroke, cerebral infarction, oedema, oxidative stress, and caspase 3 activation [48]. It may also be interesting to consider the use of immunomodulating therapy to reduce the early inflammatory response in stroke by inhibiting IL-6. The effectiveness of this form of therapy with the use of an antibody inhibiting the IL-6 receptor, tocilizumab, has already been observed in the treatment of certain diseases [49].

## 4. Materials and Methods

### 4.1. Study Group

This study involved 125 patients with acute ischemic stroke and intravenous indications for the thrombolytic treatment, aged 57 to 81 years, who were admitted to the Department of Neurology of the University Hospital in Bydgoszcz.

Patients undergoing combination therapy (intravenous and intra-arterial thrombolysis) who were diagnosed with infections, acute coronary episodes, neoplastic diseases, liver, kidney, and heart damage, or other invasive procedures before study qualification were excluded from the study. Additionally, patients with a history of stroke, chronic inflammatory and autoimmune diseases, steroids treatments, and severe injuries confirmed within ± 3 days of admission to the hospital were also excepted from this research. The local ethics committee approved the protocol of the study (nr KB 637/2016).

The Department of Neurology of the University Hospital in Bydgoszcz is known as the stroke unit, operating following Polish national norms.

A stroke was diagnosed according to the ICD 10 criteria and was confirmed at discharge based on the clinical evaluation and neuroimaging by computed tomography (CT) performed at 22–36 h after IV thrombolysis. In selected cases, additional CT scans or MRIs were performed to confirm the diagnosis. Stroke classification was carried out according to the TOAST (trial of ORG 10172 in acute stroke treatment) criteria, based on the vascular territory of the stroke. Coexisting diseases were diagnosed according to the current guidelines: hypertension with the recommendations of the European Society of Cardiology (ESC); diabetes according to the criteria of the Diabetes Association; dyslipidaemia according to ESC recommendations, and impaired renal function with eGFR < 60 mL/min/1.73 m^2^. Hyperuricemia was diagnosed with a serum uric acid concentration >6.0 mg/dL. The degrees of stenosis of the common carotid and/or internal carotid artery were assessed according to the criteria of the Polish Section of Neurosonology.

The control group consisted of 28 people aged 49–63, who were considered healthy based on the medical history and clinical evaluation. The study involved people who were not diagnosed with cardiovascular diseases, chronic inflammatory and autoimmune diseases, as well as cancer. Patients using steroids or other drugs, with infections and other diseases that could interfere with the final outcomes, were excluded from participating in the study. The consent to their participation in this research was voluntary.

### 4.2. Biochemical Testing

All biochemical analytes were routinely collected upon admission to the Department of Neurology. Samples for biochemical analyses were treated within <4.5 h and on day 1 and 7 after a stroke. Blood samples were collected from the posterior vein into 5 mL tubes employed with a clot activator and a gel separator. Serum blood was allowed to clot, and it was further centrifuged (3000 g for 15 min) and aliquoted into Eppendorf tubes. Samples were stored at a temperature of −80 °C until biochemical analysis was performed. The research protocol was approved by the bioethics committee (No. KB 637/2016). A panel of inflammatory cytokines, i.e., IL-6 and TNF-α, was estimated in serum samples collected from acute ischemic stroke patients and in the serum of healthy people. They were measured by ELISA using the commercial Diaclone kit, Besancon Cedex, France, and ImmunoBiological Laboratories (IBL International GMBH, Hamburg, Germany). All analyses were performed in accordance with the manufacturer’s instructions.

### 4.3. Statistical Methods

This study was based on a retrospective analysis of data. Descriptive statistical analysis was performed with the use of the R programming language (version 4.1.2, R Foundation for Statistical Computing, Vienna, Austria).

All continuous variables were tested for a normal distribution and equality of variance.

The normality of the distribution of continuous variables was checked with the Shapiro–Wilk test. Due to the abnormality of the variables, non-parametric Mann–Whitney U tests were used for the one-dimensional analysis of continuous variables. Binary variables were compared using the chi-square test. Spearman’s rank test was used to assess the correlation. *p* values < 0.05 were considered statistically significant. The statistical analysis also used ROC curves and the optimal cut-off points of the studied biomarkers were determined.

## 5. Conclusions

In summary, a growing body of evidence supports the fact that infarct severity and infarct volume are related to the inflammatory response. The immune response during cerebral hypoxia may be a promising target for the development of new neuroprotective therapies. The connection of inflammatory biomarkers in the pathogenesis of stroke can be used for early disease diagnosis and become an important tool for modulating brain damage during and after stroke. Future studies are needed to carefully analyse anti-inflammatory strategies, even in combination with rtPA or thrombectomy, to improve the outcomes for patients suffering from acute ischemic stroke. Well-designed studies will identify links between these biomarkers and clinical findings.

Our report, however, has some limitations. The most important one is that it is a single-centre study, which has a relatively small number of respondents. The need for informed consent to additional blood sampling for cytokine determination excluded patients with aphasia or impaired consciousness. Therefore, we mainly recruited patients with mild and moderate neurological deficits. Due to the above limitations and the conflicting results of other reports, we believe that studies on the effect of IL-6 and TNF-α levels on the prognosis of thrombolytic therapy should be continued in an even larger population and a multicentre manner. They should cover the patient population, not only those with mild to moderate neurological deficits.

## Data Availability

Data available from the authors.

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
