# Peer review of "Selected Mediators of Inflammation in Patients with Acute Ischemic Stroke"

_ijms, 2022, doi:10.3390/ijms231810614_

Round 1

Reviewer 1 Report

The authors present a prospective single-center study aimed at evaluating the concentration of pro-inflammatory cytokines IL-6 and TNF-α in the blood serum of 125 patients with ischemic stroke (AIS) and to investigate their role as new biomarkers to predict functional prognosis after thrombolytic therapy. In their study, an increased concentration of inflammatory biomarkers IL-6 and TNF-α was observed during onset and at 24 hours and 7 days after the onset of stroke compared to healthy participants who acted as a control group. The authors also demonstrated that serum levels of IL-6 before the intravenous thrombolysis were lower in the subgroup of patients with a favorable functional outcome compared to the group of patients with an unfavorable functional result (mRS: 3-6 pts) assessed at admission, discharge, 3 months and one year after stroke. The study is potentially interesting, but can be improved if the following considerations are addressed:

1.In the Introduction, please change “coagulation disorders” to “hematological diseases” because cerebral ischemia may also be the presenting manifestation of hematological disorders. This is a noteworthy clinical aspect that should also be emphasized in the Introduction (see and add this reference: Expert Rev Hematol 2016; 9: 891-901).

2.It is mandatory to describe the causes of death (neurological and non-neurological) in the study sample.

3. Review the text for typographical errors (e.g., “on-set” in the Discussion, line 259).

4.The authors demonstrated lower serum concentrations of IL-6 and TNF-α in patients with lacunar stroke. The authors should add that this circumstance may possibly be due to the fact that the pathophysiology, prognosis and clinic features of lacunar strokes are different from all other cerebral infarcts (see and add this reference: Int J Mol Sci 2022; 23, 1497).

Author Response

We would like to thank the Reviewer for the valuable remarks and comments, which significantly helped us to improve our manuscript. Below we present point-by-point responses to the comments.

In the Introduction, please change “coagulation disorders” to “hematological diseases” because cerebral ischemia may also be the presenting manifestation of hematological disorders. This is a noteworthy clinical aspect that should also be emphasized in the Introduction (see and add this reference: Expert Rev Hematol 2016; 9: 891-901).

According to the suggestion of the Reviewer, the phrase “coagulation disorders” has been changed to “hematological diseases”.

The reference (Expert Rev Hematol 2016; 9: 891-901) has been added in the introduction.

It is mandatory to describe the causes of death (neurological and non-neurological) in the study sample.

As suggested by the Reviewer, we described the causes of death in patients after a stroke.

The following sentence has been added to the manuscript:

The cause of death in patients after stroke were neurological - cerebral disorders (cerebral hemorrhage, cerebral edema) and extracerebral (severe circulatory and respiratory failure, pneumonia, sepsis and other infections).

Review the text for typographical errors (e.g., “on-set” in the Discussion, line 259).

Typographical errors have been corrected in the manuscript.

The authors demonstrated lower serum concentrations of IL-6 and TNF-α in patients with lacunar stroke. The authors should add that this circumstance may possibly be due to the fact that the pathophysiology, prognosis and clinic features of lacunar strokes are different from all other cerebral infarcts (see and add this reference: Int J Mol Sci 2022; 23, 1497).

The sentence “This circumstance may possibly be due to the fact that the pathophysiology, prognosis and clinic features of lacunar strokes are different from all other cerebral infarcts” and the reference (Int J Mol Sci 2022; 23, 1497) have been added to the manuscript.

Reviewer 2 Report

The manuscript submitted for review is a multicentre study on "Selected mediators of inflammation in patients with acute ischemic stroke". In this study, the authors focused on assessing the concentration of pro-inflammatory cytokines IL-6 and TNF-α in the blood serum of patients with ischemic stroke (AIS) and to investigate their role as new markers in predicting functional prognosis after thrombolytic therapy. The topics addressed by the authors are timely and clinically relevant. The procedures were well planned and reliably described. The authors have performed a thorough analysis of the data obtained. The discussion includes an in-depth discussion of the results obtained against the background of the current state of knowledge. Conclusions are supported by experimental data.

I suggest including a graphical design of the study in the paper to present the relationships between the research groups more clearly.

Author Response

We would like to thank the Reviewer for the valuable comment, which significantly helped us to improve our manuscript.

I suggest including a graphical design of the study in the paper to present the relationships between the research groups more clearly.

A graphical study design has been shown in Scheme 1, and an overall study summary has been provided in Scheme 2.

The following sentences have been added to the manuscript:

“The study assessed the concentration of cytokines at times < 4.5 h (0), 24 h (1) and on the seventh day (2). The mean concentration of cytokines was calculated taking into account the change in the number of patients (admission, discharge, three months, one year). The general study design is illustrated in Scheme 1.”